# Efficient Adsorption of Methyl Orange on Nanoporous Carbon from Agricultural Wastes: Characterization, Kinetics, Thermodynamics, Regeneration and Adsorption Mechanism

**Yosra Raji** [1,2], **Ayoub Nadi** [1], **Marwane Rouway** [1,3,*], **Sara Jamoudi Sbai** [1,4], **Wafaa Yassine** [2], **Abdelfattah Elmahbouby** [2], **Omar Cherkaoui** [1] **and Souad Zyade** [2]

1   Laboratory for Research in Textile Materials (REMTEX), Higher School of Textile and Clothing Industries (Esith), Casablanca 20000, Morocco
2   Genie Laboratory of Materials for Environment and Valorization (GeMEV), Aïn Chock Faculty of Sciences, Hassan II University, Casablanca 20000, Morocco
3   LPMAT Laboratory, FSAC, Hassan II University, Casablanca 20000, Morocco
4   LIMAT Laboratory, FSBM, FSAC, Hassan II University, Casablanca 20000, Morocco
*   Correspondence: marwanerouway@gmail.com

**Abstract:** Nanoporous carbon derived from *Moringa oleifera* seed waste was synthesized by an original process of flash pyrolysis followed by zinc chloride impregnation. The $N_2$-adsorption–desorption results of the optimized sample revealed a BET surface area of 699.6 $m^2/g$ and a pore size of 2 nm. It was evaluated for the adsorption of a mono azo dye, methyl orange (MeO), from aqueous solution. Four isothermal models (Langmuir, Freundlich, Dubinin–Radushkevic and Temkin) were applied to fit the experimental data. The results revealed that Langmuir is the most appropriate isothermal adsorption model to describe the adsorption process ($X^2 = 1.16$); with an adsorption capacity 367.83 mg/g at 298 K, the interaction of MeO dye with the nanoporous carbon surface is a localized monolayer adsorption. The adsorption kinetics was consistent with the pseudo-second-order model and found to correlate well with the experimental data ($X^2 = 9.06$). The thermodynamic study revealed a spontaneous and endothermic adsorption process, and the substances are adsorbed in a random manner. The desorption of MeO dye from $MO_{C-ZnCl_2}$ by sodium hydroxide solution was achieved to a level of about 84%, and the nanoporous carbon was recycled and reused at the fifth cycle. This work demonstrates that $MO_{C-ZnCl_2}$ could be employed as an alternative to commercially available activated carbon in the removal of dyes from wastewater.

**Keywords:** *Moringa oleifera*; pyrolysis; adsorption; methyl orange; isotherm; kinetic; nonlinear regression; thermodynamic

## 1. Introduction

Dyes are used in many industrial sectors such as textile dyes, paper dyes, leather dyes and in the food and cosmetic industries. The world production of dyes is estimated at more than 800,000 tons/year [1]. Among the many families of synthetic dyes, water-soluble azo dyes are the most widely used (60–70%) [1,2]. They are very stable and not biodegradable [2], mainly due to the presence of aromatic rings in their molecules. Wastewater from dye baths containing dyes of this type is sometimes discharged directly into the aquatic environment [3]. They represent a real danger for the environment; they are toxic substances, are persistent and sometimes have a mutagenic and carcinogenic effect [3,4]. It is therefore necessary to limit these pollutants as much as possible by setting up a suitable treatment method such as a discoloration unit. The treatment of wastewater containing these types of pollutants can be achieved by physical–chemical techniques [4], such as coagulation and flocculation, membrane filtration, chemical oxidation, ozonation, ion exchange and adsorption. Adsorption techniques have received considerable attention

because of their cost-effectiveness and high efficiency in removing contaminants from wastewater. Research has tended to focus on the use of biodegradable, inexpensive, locally available adsorbents from natural sources. In recent years, biochars synthesized from agricultural residues have been widely used as adsorbents to treat colored effluents due to their high porosity, high specific surface area and high adsorption capacity [5]. These biochars can be prepared by carbonization of any material containing a high proportion of carbon, such as wood [6], almond shells [7], coconut shells [8] or olive waste [9], by a high-temperature and long-duration pyrolysis treatment followed by chemical activation, or simultaneously with chemical activation [10]. *Moringa oleifera* is an abundant plant species in tropical and subtropical regions. Various parts of *Moringa oleifera* (seed, wood, bark and leaves) [11] are considered promising, biodegradable and a low-cost alternative for the production of environmentally friendly biochar for wastewater treatment.

Methyl orange (MeO) is one of the most widely used hazardous anionic azo dyes; its existence in an aquatic environment presents harmful effects to wildlife. Different methods can be used to remove the MeO dye from solutions, for example, advanced oxidation processes [12], photocatalytic degradation [13], polymer-enhanced ultrafiltration [14] and electrochemical degradation [15]. However, the above processes have disadvantages such as complexity, high operating cost and unit footprint. Adsorption is a superior technique to these other MeO mitigation methods because of its simple design and operation, low sensitivity to toxic substances and low operating costs.

For the present study, *Moringa oleifera* seed waste (husks) was used as raw precursor for nanoporous carbon preparation. These husks are one of the most readily available biomasses and are generally treated as waste. The nanoporous carbon was prepared by the original method of short-duration flash thermal pyrolysis [16], followed by zinc chloride ($ZnCl_2$) impregnation. The efficiency of the nanoporous carbon was tested for removing the anionic dye MeO. The morphological characterization of the prepared nanoporous carbon was thoroughly investigated; isothermal, kinetic and thermodynamic studies as well as the impacts of pH and temperature on the adsorption systems were comprehensively evaluated in the present study. Subsequently, the desorption of the MeO dye from the nanoporous carbon was investigated to examine the possibility of reusing it in future cycles.

## 2. Materials and Methods

### 2.1. Materials

*Moringa oleifera* seeds were collected from an agricultural farm in the city of Kenitra, located in Morocco. Zinc chloride $ZnCl_2$ (7646-85-7) and sodium thiosulfate $Na_2S_2O_3$ (10102-17-7) were provided by Sigma-Aldrich. Methyl orange dye $C_{14}H_{14}N_3NaO_3S$ (547-58-0) was purchased from MP Biomedicals. The chemical structure and physicochemical characteristics of methyl orange are presented in the Supplementary Materials in Figure S1 and Table S1, respectively.

### 2.2. Adsorbent Preparation

The seed waste $MO_R$ samples were dried at 60 °C, cut into small pieces, crushed and sieved using a 0.250 mm mesh screen. The preparation process of nanoporous carbon consists of two steps: carbonization and impregnation in the chemical agent. The carbonization process was carried out by pyrolysis with thermal shock in a laboratory tube furnace preheated to different temperatures (700, 800 and 900 °C) under atmospheric pressure for a duration time interval of between 5 and 10 min, and the results are considered as the carbonized form of *Moringa oleifera* ($MO_C$). Then, the nanoporous carbon in the optimal carbonization condition was impregnated in the agent $ZnCl_2$ at room temperature 25 ± 1 °C, [17], and at different nanoporous carbon/chemical agent mass ratios (1:1; 1:2; 1:3) for 3 h. After the impregnation process, the nanoporous carbon ($MO_{C\text{-}ZnCl_2}$), was washed with HCl (0.1 mol·L$^{-1}$) and then with distilled water until the pH was neutral. The prepared nanoporous carbon was stored in a desiccator at room temperature before

being used for the adsorption experiments. The mass yields of the $MO_C$ and $MO_{C\text{-}ZnCl_2}$ nanoporous carbons are given by Equations (1) and (2), respectively.

$$\text{Yield } (\%) = \frac{\text{Final mass } MO_C}{\text{Initial mass } MO_R} \times 100 \tag{1}$$

$$\text{Yield } (\%) = \frac{\text{Final mass } MO_{C\text{-}ZnCl_2}}{\text{Initial mass } MO_R} \times 100 \tag{2}$$

### 2.3. Proximate Analysis

The proximate analysis was measured according to the ASTM standard (American Society for Testing and Materials). $m_1$ = sample mass before heating (g), and $m_2$ = sample mass after heating (g).

- Ash: the total ash was determined by the difference in sample mass before and after heating in an electric oven at 730 °C for 8 h. The ash content is given by Equation (3).

$$\text{Ash } (\%) = \frac{m_1 - m_2}{m_1} \times 100 \tag{3}$$

- Volatile matter: the sample was deposited in a crucible closed by a lid and placed in an oven at 950 °C for 7 min. The percentage of volatile matter is calculated according to Equation (4).

$$\text{Volatile matter } (\%) = \frac{m_1 - m_2}{m_1} \times 100 \tag{4}$$

- Fixed carbon content: this refers to the non-volatile solid fraction resulting from the volatile matter and ash test, as defined by the ASTM and calculated by Equation (5).

$$\text{Fixed carbon } (\%) = 100 - (\text{Ash } (\%) + \text{Volatile matter } (\%)) \tag{5}$$

### 2.4. Iodine Index

The iodine value is a measure of the micropore content in carbon (0–10 Å), calculated using a procedure recommended by the European Chemical Industry Council (CEFIC). The blank test consists of dosing 10 mL iodine solution (0.1 N) with sodium thiosulfate solution (0.1 N) in the presence of a starch solution as an indicator until the color disappears. Then, the sample test consists of shaking 15 mL iodine (0.1 N) solution and 50 mg of each adsorbent for 4 min; 10 mL of this filtrate is dosed with sodium thiosulfate (0.1 N) solution in the presence of two drops of a starch solution. The iodine value is calculated by Equation (6).

$$I_{(\text{iodine})} \; (\text{mg/g}) = \frac{(V_B - V_S) \times N \times (126,9) \times \left( \frac{15}{10} \right)}{m} \tag{6}$$

$V_B$ and $V_S$ are the volumes of sodium thiosulfate solution required for the blank and sample titrations, respectively (mL). $N$ is the normality of the sodium thiosulfate solution in (mol/L), 126.9 is the atomic mass of iodine, and $m$ is the adsorbent mass in (g).

### 2.5. Characterization

2.5.1. FTIR

Fourier transform infrared (FTIR (Madison, WI 53711, Cleveland, OH, USA) spectroscopic analysis was used on the study surfaces to determine the surface groupings or acid–base content of the biomass and the nanoporous carbon prepared. The infrared absorption spectrum was captured between 400 and 4000 cm$^{-1}$ on a Nicolet IS10 FTIR-ATR spectrophotometer.

### 2.5.2. pHpzc

The zero-charge point pH (pHpzc) of the nanoporous carbon was determined by the standard method. Solutions of 0.01 M NaCl (50 mL) placed in separate Erlenmeyer flasks were adjusted to pH values of 2, 4, 6, 8, 10 and 12 by adding 0.1 M (HCl) or 0.1 M (NaOH). Then, 0.15 g of nanoporous carbon was added to each solution and stirred for 48 h at 25 °C. The final pH of each solution was then determined. By graphical determination, the pHpzc is the point where the curve of the final pH versus the initial pH intercepts the line.

$$\text{Final pH} = \text{Initial pH}.$$

### 2.5.3. SEM/EDS

Evaluation of nanoporous carbon morphology was performed using a Thermo Scientific Quattro S scanning electron microscope (SEM) equipped with an X-ray dispersive spectroscopy (EDS) microanalyzer.

### 2.5.4. BET

The $N_2$ adsorption–desorption isotherms of nanoporous carbons are measured using an automated sorptometer (Micrometrics ASAP 2020 Rigaku, Japan) at liquid nitrogen temperature (77.147 K). The adsorption isotherm of the equation Q (ads) = f (P/P0) is obtained by measuring the quantity of adsorbed gases for increasing values of relative pressure represented by P/P0. For each analysis, 0.2 g of sample was used. The accuracy of the measurements made by this equipment is ±3%. After degassing the samples at 200 °C under a vacuum of 550 µmHg for 11 h, the Brunauer–Emmett–Teller (BET) theory was used for the analysis of the specific surface [18], and the Barrett–Joyner–Halenda (BJH) theory was used for the analysis of the porosity based on the Kelvin equation [18].

### 2.6. Adsorption Study

In this work, the selection of appropriate models related to the adsorption of the MeO dye onto the $MO_{C\text{-}ZnCl_2}$ developed nanoporous carbon phase was performed by comparison according to nonlinear regression. Nonlinear regression is a powerful tool for analyzing scientific data, especially if the data must be transformed to fit a linear regression. The objective of nonlinear regression is to fit a model to the data under study while avoiding the experimental error fallacy of linear modeling. A program (Origin Lab 2018) allows finding the best-fit values of the model variables, of which scientific interpretations can be made.

### 2.6.1. Equilibrium Study

Adsorption of MeO dye onto $MO_{C\text{-}ZnCl_2}$ was performed using the batch method. Adsorption tests were performed at a constant temperature of 25 ± 1 °C, and at a constant volume of MeO (10 mL at the initial concentration of 500 mg/L). The effects of optional parameters were studied in order to determine the optimal conditions ensuring maximum adsorption capacity. Nanoporous carbon masses were varied between 10 and 250 mg, pH values were adjusted between 4 and 10 using HCl and NaOH (0.1 N) solutions, and stirring speeds ranged from 100 to 700 rpm. The adsorption experiments were held for 3 h to establish equilibrium conditions. The residual concentrations of the dye solutions were calculated through measuring the absorbance at a wavelength of 463 nm using the UV-visible Evolution™ 300 spectrophotometer () Sensititre™; Thermo Fisher Scientific, Cleveland, OH, USA). The quantity of MeO dye adsorbed at equilibrium was calculated using the Equation (7).

$$Q_e \ (\text{mg/g}) = \frac{(C_i - Ce)}{m} \ v \tag{7}$$

$Q_{e:}$ is the MeO quantity adsorbed by one unit mass of adsorbent at equilibrium in mg/g.

$C_i$ is the initial concentration of MeO, and $C_e$ is the equilibrium concentration of MeO in mg/L.

### 2.6.2. Kinetics Study

Kinetic studies provide information on the optimum time of adsorption, the possible rate steps and the sorption mechanism. The adsorption of MeO dye on $MO_{C\text{-}ZnCl_2}$ nanoporous carbon was studied in Erlenmeyer flasks containing 0.01 g of nanoporous carbon and 10 mL of the dye solution at an initial concentration of 500 mg/g. The mixture was stirred at a rotational speed of 300 rpm and at room temperature, 25 °C, for time intervals between 5 and 340 min; the pH was adjusted to 5 with a solution of hydrochloric acid, HCl (0.1 N). The concentration evolution of the filtrates at the different contact times was followed at a wavelength of 463 nm. The quantity of MeO dye adsorbed in milligrams onto grams of nanoporous carbon at time t (mg/g) is shown in Equation (8).

$$Q_t \ (\text{mg/g}) = \frac{(C_i - C_t)}{m} \ v \tag{8}$$

$v$ is the volume of the solution (L), $C_i$ is the initial concentration of MeO in the solution (mg/L), $C_t$ is the concentration of MeO at time t (mg/L), and $m$ is the adsorbent mass (g).

From the experimental data, the kinetic constants were determined from the application of the pseudo-first-order and pseudo-second-order models, introduced initially by Lagergren [19] and based on Equations (9) and (10), respectively.

$$Q \ (t) \ = Q_e \ (1 - e^{-K_F t}) \tag{9}$$

$K_F$ is the adsorption rate constant of the first order model $(\text{min}^{-1})$

$$Q \ (t) \ = \ Q_e \ \frac{K_S \ t}{1 + \ K_S \ t} \tag{10}$$

$K_S$ is the adsorption rate constant of the second-order model (g/mg min).

The kinetic results were analyzed using the intraparticle diffusion model based on the theory proposed by Weber and Morris and expressed by Equation (11) to identify the diffusion mechanism.

$$Q_t = K_I \ t^{1/2} + C \tag{11}$$

$K_I$ $(\text{mg/g min}^{1/2})$ is the constant of the intraparticle diffusion rate, and C (mg/g) is the constant related to the diffusion resistance.

### 2.6.3. Isotherm Study

The adsorption isotherm describes the non-kinetic relationship between the adsorbent quantity $Q_e$ and the adsorbate concentration in solution $C_e$ at a constant temperature when the adsorption process reaches equilibrium. Adsorption isotherms were obtained for 0.01 g of nanoporous carbon by varying the concentration of MeO between 500 and 1000 mg·$L^{-1}$. Adsorption isotherms were studied in closed Erlenmeyer flasks containing 10 mL dye solution at 25 °C under stirring at 300 rpm at pH = 5. The contact time was set at 3 h to ensure the establishment of an adsorption equilibrium. The quantity of MeO dye adsorbed in milligrams onto grams of nanoporous carbon at equilibrium is shown in Equation (6) above. In this context, four different models are modeled using nonlinear regression by the Origin 2018$^{®}$ software. Equations (12)–(15) present the Langmuir, Freundlich, Temkin and Dubinin–Radushkevich equations, respectively [20].

$$Q_e = \frac{Q_m \ K_L C_e}{1 + K_L C_e} \tag{12}$$

($K_L$ (L/mg) is the constant of the Langmuir isotherm, and $Q_m$ (mg/g) is the maximum quantity adsorbed.)

$$Q_e = K_F \, C_e^{1/n} \tag{13}$$

$K_F$ (L/mg) is the isothermal constant of Freundlich, and $1/n$ is the heterogeneity factor.

$$Q_e = \frac{RT}{B_T} \ln (K_T \, C_e) \tag{14}$$

($B_T$ and $K_T$ are the Temkin isotherm constants, while R is the universal gas constant (0.008314 kJ/(mol K).)

$$Q_e = Q_m \, e^{(-B_D \, (RT \, \ln (1+1/C_e))2)} \tag{15}$$

The value of $B_D$ is related to the sorption energy E, and $Q_m$ (mg/g) is the maximum quantity adsorbed.

### 2.6.4. Statistical Study

The choice of the model is a scientific decision and should not be based solely on the shape of the graph. Several parameters were used to determine the appropriate model. The verification of this proposed model's adequacy to describe the adsorption kinetics and adsorption isotherms is performed with the values determination coefficient ($R^2$), adjusted coefficient determination ($R^2_{adjusted}$) and chi-square error function ($X^2$) using Equation (16) [20].

$$X^2 = \sum_{i=1}^{n} \frac{\left(Qe_{cal} - Qe_{exp}\right)2}{Q_{exp}} \tag{16}$$

$Qe_{cal}$ is the quantity of MeO adsorbed at the calculated equilibrium (mg/g), and $Qe_{exp}$ is the quantity of MeO adsorbed at the experimental equilibrium (mg/g).

### 2.7. Thermodynamic Study

Thermodynamic parameters are used to reveal the energy changes that occur in the adsorption process. Thermodynamic studies were performed over temperatures ranging from 298 to 323 K. The changes in thermodynamic parameters like standard free energy ($\Delta G°$), enthalpy ($\Delta H°$) and entropy ($\Delta S°$) in the adsorption process were determined by plotting Ln $K_D$ as a function of $1/T$ and using the thermodynamic Equations (17)–(19).

$$\ln K_D = \frac{\Delta S°}{R} - \frac{\Delta H°}{RT} \tag{17}$$

$K_D = Q_e/C_e$ is the distribution coefficient, while R is the universal gas constant (0.008314 (K J/mol).

$$\Delta G° = -RT \ln K_D \tag{18}$$

$$\Delta G° = \Delta H - T\Delta S° \tag{19}$$

### 2.8. Desorption and Regeneration

For saturation conditions, 0.01 g of nanoporous carbon was introduced to 10 mL of MeO dye solution at 500 mg/g for 3 h at pH = 5, at room temperature and under stirring at 300 rpm. In order to examine the power of some parameters to desorb the dye, the nanoporous carbon loaded with the dye was removed from the adsorption solution by filtration on Whatman filter paper and washed with distilled water to remove traces of unadsorbed MeO. We then immersed it in 10 mL of an alkaline desorption solution of NaOH (0.01 N) for 2 h at 25 °C under stirring at 700 rpm. After the desorption test, the concentration of MeO in this solution was measured using an Evolution™ 300 spectrophotometer (Thermo Fisher Scientific, USA) at wavelength 463 nm. The nanoporous carbon obtained from the desorption solution was recovered by filtration, washed with

distilled water, neutralized, dried and reused for future adsorption–desorption cycles. The desorption efficiency is given by Equation (20).

$$\text{Desorption }(\%) = \frac{Q_{DES}}{Q_{ADS}} \times 100 \tag{20}$$

$Q_{DES}$ and $Q_{ADS}$ represent the quantities of MeO desorbed and adsorbed in mg/g, respectively.

## 3. Results

### 3.1. Proximate Analysis, Iodine Index and Characterization

Table 1A presents the results of the ash, volatile matter and fixed carbon contents for the $MO_R$ biomass and $MO_C$ nanoporous carbons at different pyrolysis temperatures (700, 800 and 900 °C) and at a time interval between 5 and 10 min. It can be seen that the ash content is quite low (5.7% for $MO_R$ and 2.01% for $MO_C$); these small values can be explained by the fact that the nanoporous carbons are prepared from a plant material richer in carbon than in mineral matter, which suggests a high purity for the nanoporous carbon produced. The volatile matter decreases with increasing pyrolysis temperature in the order $MO_R > MO_{CA\ 700°C} > MO_{CA\ 800°C} > MO_{CA\ 900°C}$. On the other hand, in the high-temperature pyrolysis phase, the volatile substances become unstable due to the heat and are released as gaseous and liquid products. The fixed carbon content expresses the actual quantity of pure carbon remaining after complete decomposition. It increases with the pyrolysis temperature, and a very high rate (85.1%) was obtained at 900 °C. It is noted that at higher temperatures, a slight decrease in yield was observed, which is in agreement with other works [21]. The pyrolysis time did not influence the proximate analysis much, which allowed us to delineate that the optimum time for heat-shock pyrolysis is 5 min at a temperature of 900 °C. It can also be seen that the carbonization and impregnation in the chemical agent $ZnCl_2$ greatly increased the specific surface area in the order of $MO_R < MO_{CA} < MO_{C-ZnCl_2}$ (Table 1A,B), and the 1:2 $ZnCl_2$ impregnation ratio is maintained as the optimum to form a large specific surface area (Table 1B). Thus, the effect of carbonization is to enrich the material with carbon and create the first pores, giving a greater specific surface area, and subsequently, the subsequent development of the specific surface area increases by the effect of chemical $ZnCl_2$ impregnation to create new pores or enlarge the specific surface area considerably [22]. According to the analysis of the iodine index, the nanoporous carbon obtained from the optimal mode of $MO_{C-ZnCl_2}$ would be able to store 611.29 mg/g of iodine ($I_2$). We note that the treatment of the *Moringa oleifera* husks by transforming them into nanoporous carbon makes their specific surface evolve, giving them a high microporosity and an interesting adsorbent power. Let us note that this index is linked to the number of micropores whose diameters are lower than 10 Å available in the structure of the analyzed nanoporous carbon.

The infrared analysis spectra of the raw material and the prepared $MO_{C-ZnCl_2}$ nanoporous carbon are shown in Figure 1A. The spectrum of $MO_R$ before carbonization and $ZnCl_2$ impregnation shows distinct bands; the broad absorption band at 3400 cm$^{-1}$ corresponds to hydrogen elongation vibrations of O-H hydroxyl groups [23] (of carboxyls, phenols or alcohols) and adsorbed water. It also corresponds to elongation vibrations of O-H of cellulose, pectin and lignin [24]. The bands between 2917 and 2885 cm$^{-1}$ result mainly from the vibrations of elongation of C-H of the aliphatic molecules [25]. The small band around 1645 cm$^{-1}$ is attributed to the elongation vibrations of the C=O groups [26] of the ketones and aldehydes present in the lignin. The band at 1503 cm$^{-1}$ is attributed to the elongation vibrations of the C=C bonds. The bands between 1000 and 1039 cm$^{-1}$ are assigned to the vibrations of the C-O bonds present in the lignocellulose. As can be observed, the spectrum of the $MO_{CA-ZnCl_2}$ nanoporous carbon shows fewer absorption bands than the spectrum of the $MO_R$ biomass, indicating that some functional groups present in the biomass disappeared after the pyrolysis at high temperature. The band around 3280 cm$^{-1}$ related to the presence of hydroxyl became weaker due to the loss of moisture and water of hydration following the increase in pyrolysis temperature. The band

attributed to C=C was decreased, which confirms the creation of new sites by the chemical agent. The appearance of a ZnO band after impregnation in ZnCl$_2$ at 370 cm$^{-1}$ is due to the doping of Zn$^{2+}$ ions in nanoporous carbon matrix [27].

**Table 1.** **A**: Proximate analysis, Surface BET and Iodine Index of MOR and MOC at different temperatures and pyrolysis times; **B**: Yield, Surface BET and Iodine Index of MOC-ZnCl$_2$ at different impregnation rates MOC/ZnCl$_2$.

| A | | | | | | |
|---|---|---|---|---|---|---|
| **Adsorbent** | **Yield %** | **Volatile %** | **Ash Content %** | **Carbon Fixed %** | **Surface BET (m$^2$/g)** | **Iodine Index mg/g** |
| MO$_R$ | —— | 60.983 | 5.702 | 33.315 | 92.155 | 77.961 |
| MO$_C$ | | | | | | |
| T °C · Pyrolysis time | | | | | | |
| 700 °C · 5 min | 47.196 | 30.691 | 2.126 | 67.183 | 216.525 | 260.786 |
| 700 °C · 10 min | 45.110 | 29.255 | 2.080 | 68.665 | | |
| 800 °C · 5 min | 43.114 | 20.962 | 2.054 | 76.984 | 369.029 | 369.656 |
| 800 °C · 10 min | 42.931 | 20.159 | 2.036 | 77.805 | | |
| 900 °C · 5 min | 41.360 | 12.895 | 2.019 | 85.086 | 422.634 | 420.370 |
| 900 °C · 10 min | 40.169 | 12.113 | 2.012 | 85.875 | | |

| B | | | |
|---|---|---|---|
| **MO$_{C-ZnCl_2}$** | | | |
| **Impregnation Rate MO$_C$/ZnCl$_2$** | **Yield %** | **Surface BET (m$^2$/g)** | **Iodine Index mg/g** |
| 1:1 | 40.964 | 625.222 | 589.854 |
| 1:2 | 40.120 | 699.696 | 611.291 |
| 1:3 | 40.033 | 602.005 | 589.855 |

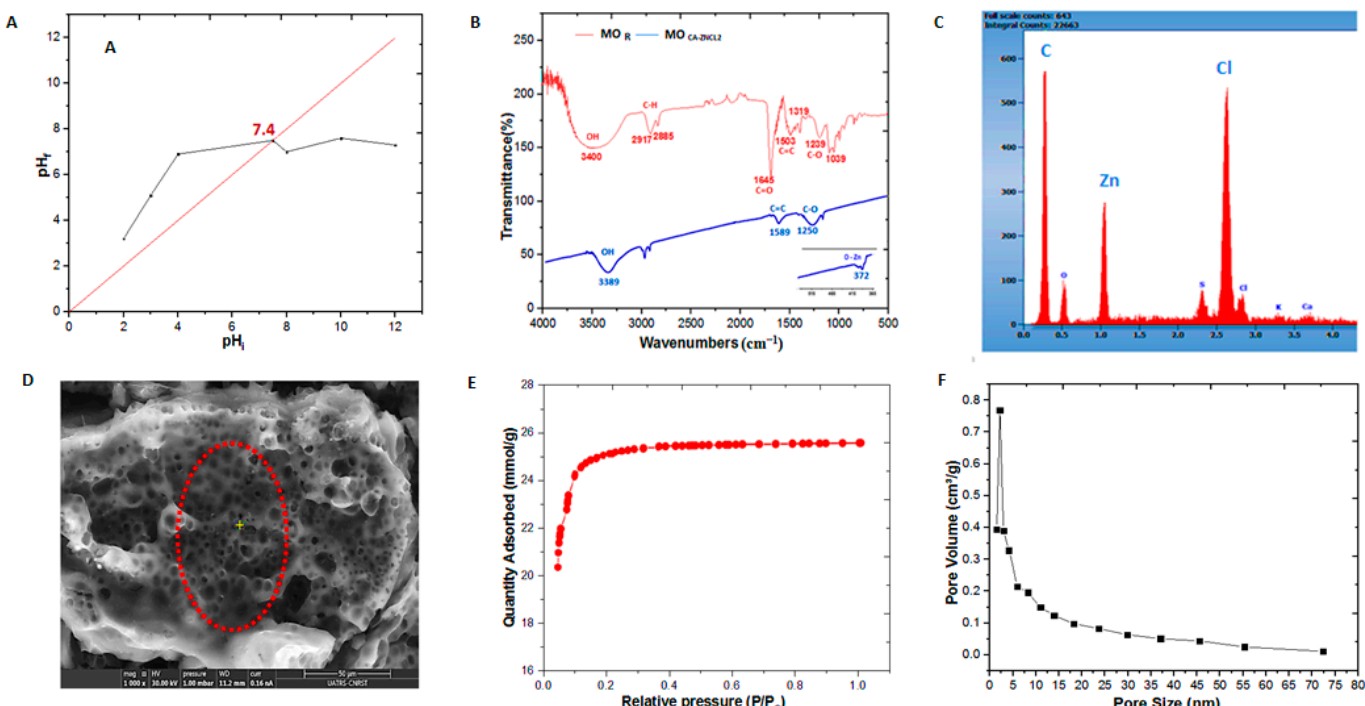

**Figure 1.** MO$_{C-ZnCl_2}$ characterization: (**A**) pHpzc, (**B**) FTIR, (**C**) EDS, (**D**) SEM, (**E**) Adsorption/desorption isotherms of N$_2$/77 K, (**F**) Pore distribution by BJH method.

The determination of the pHpzc of the MO$_{C-ZnCl_2}$ nanoporous carbon from the evolution of the final pH as a function of the initial pH is presented in Figure 1B. The point of

zero charge is close to pH = 7.4; this nanoporous carbon has a negative surface at pH > 7.4 and a positive one at pH < 7.4.

Figure 1C shows the morphology of $MO_{C-ZnCl_2}$ nanoporous carbon particles studied via scanning electron microscopy (SEM). The picture shows that the nanoporous carbon has a highly developed porosity containing pores separated by thin walls, which could be useful for the storage of organic pollutants. Referring to the results of EDS analysis presented in Figure 1D, we note the strong dominance of carbon atoms in the molecular composition of $MO_{C-ZnCl_2}$, proving its high purity during the pyrolysis process, and also the existence of relatively high chlorine and zinc contents, proving the success of $ZnCl_2$ impregnation. From these observations, we distinguish the high quality of the prepared nanoporous carbon.

Figure 1E shows the nitrogen adsorption–desorption isotherms at 77.147 K of the $MO_{C-ZnCl_2}$ nanoporous carbon. According to the IUCPA classification, the curve illustrates a type I (b) isotherm characterized by a large adsorption at very low relative pressure (P/P0 < 0.1) and a long plateau that extends to P/P0 $\approx$ 1.0. Type I (b) isotherm materials have wider micropores [28]. This is confirmed by the pore distribution curve obtained by the BJH method [18] (Figure 1F), where it can be observed that the $MO_{C-ZnCl_2}$ nanoporous carbon mainly produces pores with sizes ranging from $\approx$2 nm, which indicates the presence of micropores [28] in the surface of the $MO_{CA-ZnCl_2}$ nanoporous carbon. The microporous structure produced by the chemical agent $ZnCl_2$ is also noted by Mahmoudi [29] and by Marcela [30]. The results of the textural properties are shown in Table S2 in the Supplementary Materials. The external surface area comprises 97.696 m$^2$/g (13% of the specific surface area), while the micropore surface area comprises 601.99 m$^2$/g (86% of the specific surface area), which is favorable for the adsorption of small-sized pollutants.

### 3.2. Adsorption Study

The effect of adsorbent mass on the adsorption process is illustrated in Figure 2A; the dye quantity adsorbed decreases with increasing nanoporous carbon mass. However, at high masses, the adsorbent particles were competing with each other to bind the same dye molecules [31], resulting in the formation of agglomerations of $MO_{C-ZnCl_2}$ particles in the solution; 10 mg of nanoporous carbon is defined as the optimal condition for the follow-up study. The effect of pH on the adsorption of MeO is shown in Figure 2B. It can be explained based on the pHpzc (Figure 1A); the pHpzc of $MO_{C-ZnCl_2}$ is 7.4. At pH < pHpzc, the adsorbent surface is positively charged due to the high concentration of $H_3O^+$ protons in solution, which creates an attractive force between the anionic ions of the MeO dye and the sites on the adsorbent surface (favorable adsorption at pH = 5). The agitation speed which allows obtaining a better homogeneity of the mixture is obtained at 300 tr/min (Figure 2C). It can be seen that below 300 rpm, the adsorption is reduced due to the internal propagation slowing down the adsorption, and above 300 rpm the desorption process starts to appear.

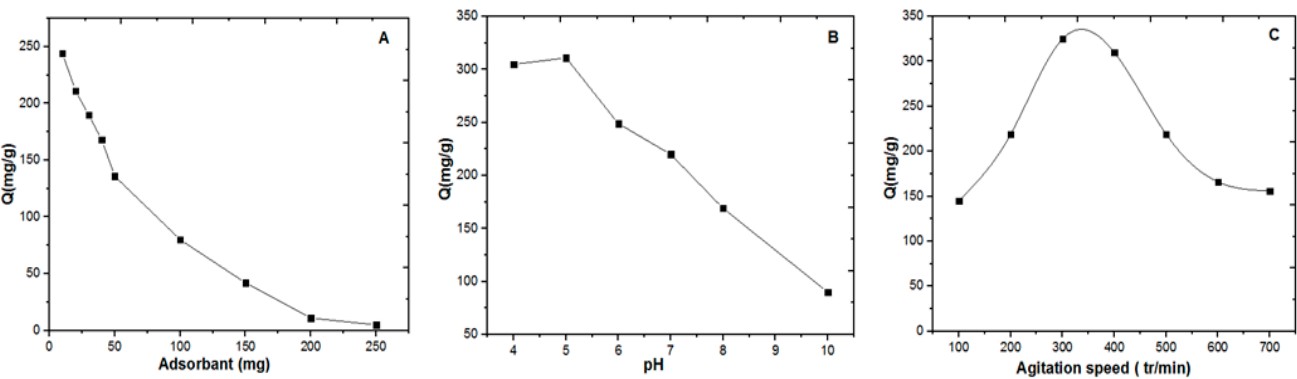

**Figure 2.** Evolution of the adsorption capacity of MeO dye as a function of: (**A**) adsorbent mass, (**B**) pH, (**C**) Agitation speed.

The kinetic study was performed by using the prepared $MO_{C\text{-}ZnCl_2}$ nanoporous carbon to adsorb the MeO dye at an initial concentration of 500 mg/L. Figure 3A shows that the contact time required to reach adsorption equilibrium is about 125 min, with an adsorbed quantity of about 326 mg/g. Adsorption begins quickly until 25 min into the reaction; then, the kinetics become increasingly slower until equilibrium is established. This can be explained by the fact that at the beginning of the adsorption, the number of active sites available on the adsorbent surface is much larger than the number of sites remaining after a certain time [27]. For high contact times, the molecule needs some time for diffusion inside the adsorbent pore. The residual unadsorbed quantity is explained by the adsorbent surface saturation (all adsorption sites are occupied). The second-order model is found to be the best for MeO dye adsorption on $MO_{C\text{-}ZnCl_2}$, as expressed by the good agreement between theoretical and experimental values of $Q_e$, as well as the model being characterized by the determination coefficients and adjusted determination coefficients closest to unity ($R^2 = 0.98$) and the small function value of $X^2 = 9.06$ (Table 2). Validation of this model indicates that the adsorption capacity is only related to the number of suitable sites available on the surface. Analysis of kinetic data by other researchers also showed that the pseudo-secondary rate equation can simulate the adsorption of methyl orange dye (MeO) on nanoporous carbon with good agreement [32–34].

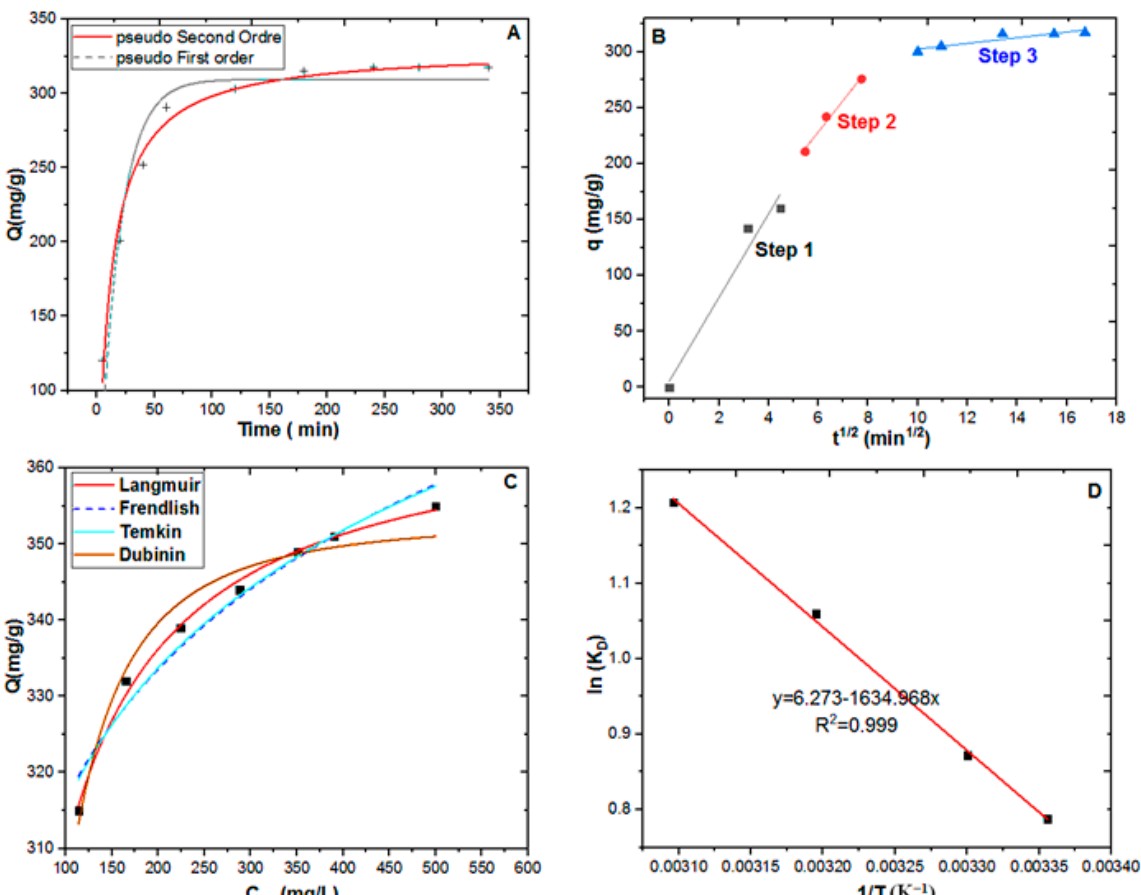

**Figure 3.** Modeling of adsorption of MeO on $MO_{C\text{-}ZnCl_2}$. (**A**) Kinetics: PFO and PSO, (**B**) Intraparticle diffusion, (**C**) Isotherm, (**D**) Representation of the Van't Hoff line.

**Table 2.** Pseudo-first-order and pseudo-second-order parameters of MeO adsorption on $MO_{C-ZnCl_2}$.

| $C_0$ (mg·L$^{-1}$) | $Qe_{exp}$ (mg g$^{-1}$) | Model PFO | | | | | Model PSO | | | | |
|---|---|---|---|---|---|---|---|---|---|---|---|
| | | $K_F$ (min$^{-1}$) | $Q_{e\ th}$ (mg g$^{-1}$) | $R^2$ | $R^2_{ajusted}$ | $\chi^2_{Réduced}$ | $K_S \cdot 10^4$ (g mg$^{-1}$ min$^{-1}$) | $Q_{e\ th}$ (mg g$^{-1}$) | $R^2$ | $R^2_{ajusted}$ | $\chi^2_{Reduced}$ |
| 500 | 326.215 | 0.053 | 309.521 | 0.921 | 0.910 | 422.910 | 2.850 | 329.821 | 0.983 | 0.981 | 9.061 |

The plots of the intraparticle diffusion model showed that the adsorption mechanism of the MeO dye on the $MO_{C-ZnCl_2}$ nanoporous carbon has three different linear portions, indicating that three steps took place (Figure 3B). The first is attributed to the diffusion of MeO through the solution or through the boundary layer film on the outer surface of the adsorbent (high slope). The second step describes intraparticle diffusion (moderate slope) induced by adsorbate mass transfer within the adsorbent surface pores. The third step exhibits an equilibrium state due to the adhesion of the adsorbate on the adsorption site (almost zero slope) [35]. The plot of the $Q_t = f\ (t^{1/2})$ curve does not cross the origin, demonstrating that intraparticle diffusion participates in the adsorption mechanism but is not the only limiting step [36]. Table 3 shows that the diffusion constants vary in the order of $K_{I1} > K_{I2} \gg K_{I3}$, indicating that film diffusion occurs at a faster rate than intraparticle diffusion, and that the third adsorption step corresponds to the adsorption equilibrium, where MeO molecules adhere to the surface of the materials in a stationary state, as illustrated by the low values of $K_{i3}$. The intercepts C are proportional to the thickness of the boundary layer, and the effect of the boundary layer increases with the value of C. The large value of $C_3$ (277.134 mg/g) indicates that the adsorption capacity is larger.

**Table 3.** Parameters of the intraparticle diffusion model of MeO adsorption on $MO_{C-ZnCl_2}$.

| First Step | | | Second Step | | | Third Step | | |
|---|---|---|---|---|---|---|---|---|
| $K_{I1}$ (mg/g min$^{1/2}$) | $C_1$ (mg/g) | $R^2$ | $K_{I2}$ (mg/g min$^{1/2}$) | $C_2$ (mg/g) | $R^2$ | $K_{I3}$ (mg/g min$^{1/2}$) | $C_3$ (mg/g) | $R^2$ |
| 37.487 | 5.339 | 0.982 | 28.139 | 59.575 | 0.991 | 2.538 | 277.134 | 0.927 |

Figure 3C shows the equilibrium adsorbed quantity variation as a function of equilibrium concentration ($Q_e = f\ [C_e]$) by projecting the experimental values onto the isothermal models using the nonlinear method. The values of the Langmuir determination coefficients—$R^2 = 0.994$ and $R^2_{adjusted} = 0.993$—are very close to unity, and the error function $X^2 = 1.16$ is very small compared to the other values of the model (Table 4), indicating that the Langmuir model perfectly describes the adsorption process of the MeO dye. It is also evident that the value of the Langmuir separation factor $R_L = 0.018$ belongs to the validity range (between 0 and 1), indicating a very favorable adsorption of MeO substrates on $MO_{C-ZnCl_2}$ which is hardly reversible. Methyl orange was strongly adsorbed by $MO_{C-ZnCl_2}$ ($Q_{max} = 367.83$ mg/g) according to the Langmuir model, which means that the adsorption is a homogeneous monolayer, that there are no interactions between the adsorbed species and that the adsorbent sites are at the same energy level. Other researchers have also shown that the Langmuir model is adequate to describe the adsorption process of methyl orange dye on nanoporous carbon produced by finger citron residue [33], by waste cellulose [34] and by date pits [29].

**Table 4.** Parameters for modeling isotherms of MeO adsorption on $MO_{C-ZnCl_2}$.

| Isotherms | | | |
|---|---|---|---|
| **Langmuir** | | **Freundlich** | |
| $R^2$ | 0.994 | $R^2$ | 0.958 |
| $R^2_{adjusted}$ | 0.993 | $R^2_{adjusted}$ | 0.949 |
| $\chi^2_{Reduced}$ | 1.161 | $\chi^2_{Reduced}$ | 9.431 |
| $K_L$ (L·mg$^{-1}$) | 0.053 | $K_F$ (mg$^{(1-1/n)}$ L$^{(1/n)}$/g) | 222.269 |
| $Q_{max}$ (mg·g$^{-1}$) | 367.835 | $n$ | 8.494 |
| $R_L$ | 0.018 | | |
| **Temkin** | | **Dubinin–Radushkevich** | |
| $R^2$ | 0.963 | $R^2$ | 0.963 |
| $R^2_{adjusted}$ | 0.956 | $R^2_{adjusted}$ | 0.956 |
| $\chi^2_{Reduced}$ | 8.261 | $\chi^2_{Reduced}$ | 8.262 |
| $b_T$ (kJ·mol$^{-1}$) | 0.009 | $q_D$ (mg·g$^{-1}$) | 353.210 |
| $K_T$ (L·g$^{-1}$) | 18.980 | $K_D$ (kmol$^2$·J$^{-2}$) | 256.493 |

### 3.3. Thermodynamics Study

The plot of Ln $K_D$ as a function of 1/T is shown in Figure 3D. The thermodynamic parameter results obtained are grouped in Table 5. From these results, the quantity of adsorbed MeO dye ($Q_e$) increases as temperatures increase from 298 K to 323 K, giving a positive value to ΔH enthalpy (ΔH = 13.593 kJ/mol), which implies that the adsorption process is endothermic [37]. In addition, a higher temperature facilitates the adsorption, and it is also noted that the interactions enter the physisorption range because ΔH° < 20 kJ/mol. The positive value of entropy change (ΔS° = 0.052 kJ/mol·K) suggests an increased randomness of MeO molecules on the adsorbent surface due to structural changes in the adsorbate–adsorbent complex [38], confirming the affinity of the adsorbent for the dye and the non-reversible nature of the adsorption process [38]. The Gibbs free energy values ΔG° are negative, which indicates the spontaneous nature of MeO dye adsorption on $MO_{C-ZnCl_2}$ and the favorable nature of the adsorption for all temperatures studied [37]. The same thermodynamic conditions were obtained in the study of León [38] concerning the adsorption of methyl orange onto granular activated carbon [38].

**Table 5.** Representation of the thermodynamic parameters (ΔG°, ΔH° and ΔS°) obtained by Van't Hoff.

| T (K) | 1/T (K$^{-1}$) | $C_e$ (mg/L) | $Q_e$ (mg/g) | $K_D$ | Ln $K_D$ | ΔG° (kJ mol$^{-1}$) | ΔH° (kJ mol$^{-1}$) | ΔS° (kJ mol$^{-1}$ K$^{-1}$) |
|---|---|---|---|---|---|---|---|---|
| 298 | 0.00335 | 150.131 | 330.211 | 2.199 | 0.788 | −1.903 | | |
| 303 | 0.00330 | 141.665 | 339.245 | 2.394 | 0.872 | −2.163 | | |
| 313 | 0.00319 | 126.564 | 365.614 | 2.888 | 1.060 | −2.683 | 13.593 | 0.052 |
| 323 | 0.00309 | 113.325 | 378.986 | 3.344 | 1.207 | −3.203 | | |

### 3.4. Adsorption–Desorption

Regeneration is a critical factor in determining the useful life of adsorbents. In other words, it is the maximum number of times an adsorbent can be used for adsorption while retaining its adsorption capacity. The desorption experiment of MeO previously adsorbed on $MO_{C-ZnCl_2}$ nanoporous carbon was performed by contact with NaOH solution (0.01 N) as a desorption agent. The reusability and stability of the nanoporous carbon were determined by following the variations of adsorption–desorption cycles, and the results are presented in Figure 4. It was found that the desorption reached from 84% to 81% during the fifth cycle of desorption; also the nanoporous carbon could be reused up to the fifth cycle with an adsorption efficiency higher than 82%. The small reduction in adsorption capacity from one cycle to the other can be explained by electrostatic interactions that are not fully recoverable in the following cycles [39]. It can be deduced from this result that the

prepared $MO_{C-ZnCl_2}$ nanoporous carbon can be used continuously, and its efficiency does not decrease rapidly.

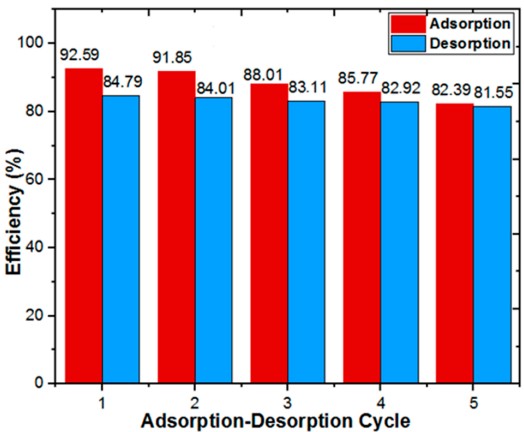

**Figure 4.** Adsorption–desorption cycle of $MO_{C-ZnCl_2}$ with alkaline solution NaOH (0.01 N).

*3.5. Treatment of the Real Textile Effluent*

In order to evaluate the application of $MO_{C-ZnCl_2}$ for the treatment of industrial textile effluents, a real MeO dyeing effluent was recovered and treated with nanoporous carbon according to the optimized adsorption conditions. The spectra of the untreated and treated solutions were recorded from 300 to 800 nm on the UV-vis spectrophotometer. The percentage of dye removed from the effluent depends on the areas under the absorption bands. It is observed that the absorbance spectrum of the real textile dyeing effluent diluted 5 times (Figure 5A) contains a peak at the wavelength 463 nm corresponding to the existence of the MeO dye. However the spectrum after treatment contains no peaks, signifying the complete discoloration of the effluent (Figure 5B). The main characteristics of the textile effluent before and after treatment are listed in Table 6. The removal of COD and turbidity reach 97% and 91%, respectively. The treated effluent shows all the characteristic properties included in the discharge limit values according to WHO standards. The prepared $MO_{C-ZnCl_2}$ nanoporous carbon proves to be an excellent adsorbent for textile effluents loaded with the dye methyl orange (MeO).

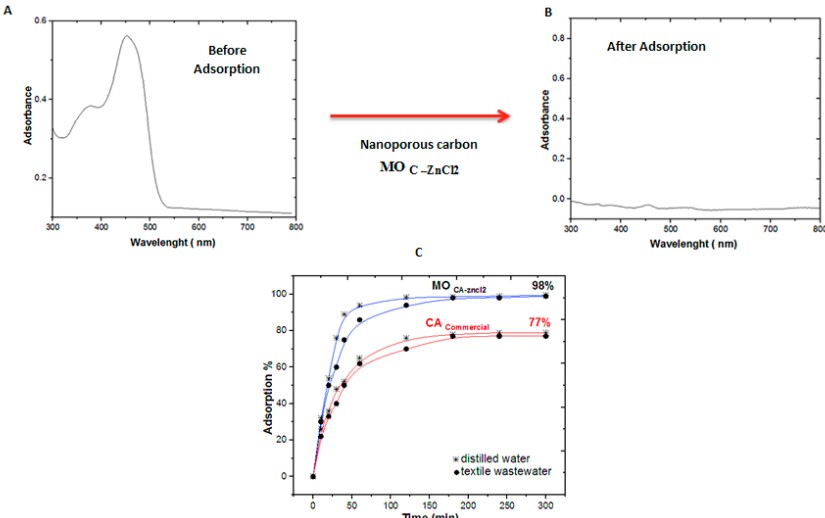

**Figure 5.** UV-vis spectrophotometer spectra of the dyeing effluent. (**A**) Before adsorption (**B**) After adsorption, (**C**) Comparative adsorption of $MO_{C-ZnCl_2}$ nanoporous carbon vs. commercial activated carbon for MeO adsorption.

**Table 6.** Characteristics of textile wastewater before and after adsorption on MO $_{C-ZnCl_2}$.

| Characteristic | Before Adsorption | After Adsorption | Removal | Norms of Limits According to the World Health Organization (WHO) for Wastewater |
|---|---|---|---|---|
| Turbidity (NTU) | 24.71 | 2.19 | 91.13% | <5 |
| COD (mg·L$^{-1}$) | 519.53 | 11.80 | 97.72% | <90 |
| BOD$_5$ (mg·L$^{-1}$) | 11.29 | 7.06 | 37.46% | <30 |
| COD/DBO$_5$ | 46.01 (Not biodegradable) | 1.67 (Easily biodegradable) | — | <3 (Biodegradable) |
| TSS (mg·L$^{-1}$) | 1.36 | 0.50 | 63.23% | <20 |
| pH | 7.52 | 6.96 | — | 6.5–8.5 |
| Temperature (°C) | 25 | 25 | — | <30 |
| Color | Light orange | Incolore | Total discoloration | Clair/incolore |
| Odor | Inodore | Inodore | — | Inodore |

A comparative experiment with a commercial activated carbon (CAS: 7440-44-0) was performed to evaluate the feasibility of the nanoporous carbon prepared in this study, MO$_{C-ZnCl_2}$. The adsorption of MeO in distilled water and textile wastewater using the two adsorbents is shown in Figure 5C. It shows that the MO$_{C-ZnCl_2}$ nanoporous carbon removed up to 98% of the initial concentration of MeO compared to the commercial activated carbon's 76.25%. Nitrogen physisorption analysis, BET, was used to understand the differences between the two adsorbents in MeO dye removal. The commercial activated carbon exhibits larger pore diameters on its surface (18 nm) compared to the nanoporous carbon prepared in this study (2 mm) (see Table S2 in Supplementary Material). Nevertheless, the commercial activated carbon contains mainly mesopores in its surface (72.99%). Therefore, they cannot retain the small-molecular-size MeO inside the pores. On the other hand, MO$_{C-ZnCl_2}$ offers high microporosity and is able to retain the dye inside. We can deduce that the MO$_{C-ZnCl_2}$ nanoporous carbon is selective to small molecules.

## 4. Discussion

The mass yield of MO$_C$ nanoporous carbon under optimal pyrolysis conditions is 41.36% (temperature of the oven preheated to 900 °C during 5 min of pyrolysis). This result compares favorably with the results of previous work by Santoset [40] and Myek [41], who found nanoporous carbon mass yields from *Moringa oleifera* ranging from 10 to 30% using flash pyrolysis at very high temperature and with a long pyrolysis time. According to our previous study [16], the original method based on the technique of direct pyrolysis with thermal shock at high temperature (oven preheated to 900 °C), with a short duration of pyrolysis and at atmospheric pressure, allows to increase the yield to ensure a fast conversion, quickly transform the biomass into carbon and avoid the formation of byproducts and emissions while optimizing the product performance [16].

It is also noted that the notable variations in the surface morphology of *Moringa oleifera* husks after carbonization are due to the degradation of the lignocellulosic material at high temperature followed by the evaporation of volatile compounds, leaving well-developed samples in pores and holes on the surface. The effect of carbonization is to enrich the material in carbon and to create the first pores, yielding a larger specific surface, and thereafter the further development of the surface depends on the strength of the chemical ZnCl$_2$ impregnation to form new pores and to enlarge considerably the specific surface. The application of ZnCl$_2$ in the chemical impregnation process generally improves the nanoporous carbon content through the formation of an aromatic graphitic structure; the chemical liquid is then intercalated into the nanoporous carbon matrix to produce new

pores [42]. Our study confirmed that zinc chloride is a powerful and aggressive activator that is able to corrode the walls of the nanoporous carbon by creating more porosity, and thus more surface area subsequently ($S_{BET}$ = 699.69 m$^2$/g). Similar studies have observed the performance of $ZnCl_2$ on the formation of porous structures of adsorbents [30,42,43].

Nanoporous carbons are materials with an amphoteric character; thus, depending on the pH of the solution, their surfaces could be positively or negatively charged. The methyl orange (MeO) dye is a water-soluble anionic dye due to its $SO_3^-$ sulfonate group. The prepared $MO_{C\text{-}ZnCl_2}$ nanoporous carbon contains polar groups such as hydroxyls and carboxyls on its surface. The electrical charge of the adsorbent depends on the pH of the medium due to the ionization of these surface functional groups. The adsorption of MeO increases with increasing positive surface charge (Figure 2B). The adsorption is most noticeable when the pH is below pHpzc = 7.4, and from pH = 5, the reaction becomes an electrostatic attraction between oppositely charged species (nanoporous carbon and dye). These assumptions let us suppose the surface considerations presented by the following chemical equations with $S_A$ representing the surface of the adsorbent:

$$\text{At pH} \leq \text{pHpzc: } S_A\text{-OH} + H_3O^+ \rightarrow S_A\text{-OH}_2^+ + H_2O$$

$$\text{At pH} = \text{pHpzc: } S_A\text{-OH}_2^+ \rightarrow S_A\text{-OH} + H^+$$

$$\text{At pH} \geq \text{pHpzc: } S_A\text{-OH} + OH^- \rightarrow S_A\text{-O}^- + H_2O$$

More than that, MeO dye was strongly adsorbed by $MO_{C\text{-}ZnCl_2}$ ($Q_{max}$ = 367.83 mg/g) due to impregnation in $ZnCl_2$, which transformed most of the oxygenated anionic sites ($C\text{-}O^-$) of the nanoporous carbon into cationic sites ($C\text{-}O\text{-}Zn^+$), resulting in a higher affinity to anionic molecules. On the other hand, methyl orange molecules can also be adsorbed by other types of π-π interactions and by hydrogen bonds [44]. We can therefore propose a mechanism as follows in Figure 6.

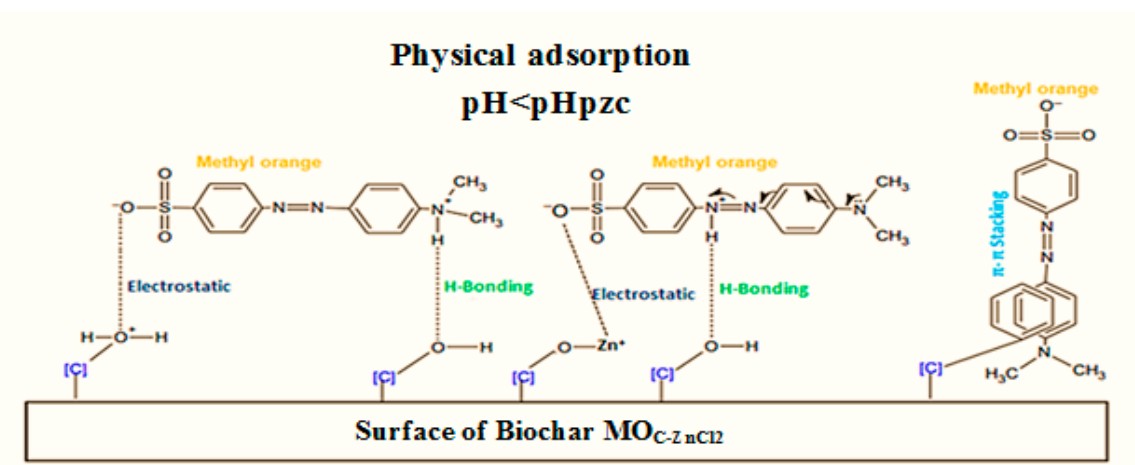

**Figure 6.** Proposed mechanism for the adsorption of MeO on $MO_{C\text{-}ZnCl_2}$.

Figure 7 represents the suggested adsorption and desorption mechanism between the $MO_{C\text{-}ZnCl_2}$ nanoporous carbon and the anionic dye MeO. Because the adsorption of MeO on $MO_{C\text{-}ZnCl_2}$ was very poor at higher pH values (Figure 2B), this implies that a basic medium could be used for the desorption process. In the present study, MeO-loaded $MO_{C\text{-}ZnCl_2}$ was regenerated using an alkaline NaOH solution (Figure 4). The effect of NaOH on the desorption of MeO dye is explained by the increase of negatively charged sites at basic pH, facilitating the desorption of MeO from the surface of $MO_{C\text{-}ZnCl_2}$ through the formation of repulsive forces. The abundance of oxide groups on the $MO_{C\text{-}ZnCl_2}$ surface was fixed by a strong chemical agent (NaOH), which supplanted the MeO molecule and thus increased

the desorption activity. Subbaiah [39] and Mittal [45] also performed a regeneration of the MeO dye from the adsorbents with an NaOH solution.

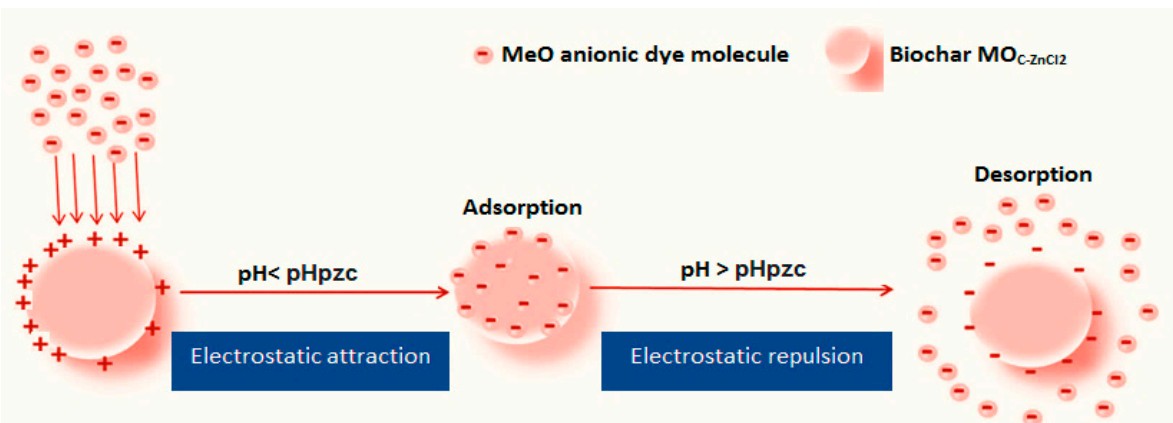

**Figure 7.** Schematic representation for adsorption and desorption of MeO anionic dye onto $MO_{C-ZnCl_2}$ nanoporous carbon.

Comparing the list of adsorption capacities of different activated carbons used for the adsorption of methyl orange dye (MeO) from aqueous solutions, the results show that the adsorption capacity of $MO_{C-ZnCl_2}$ nanoporous carbon is higher than other activated carbons presented in the literature (Table 7). In addition, the short thermal flash pyrolysis time of 5 min as well as the low mass ratio of activator to precursor used (1:2) at room temperature in this study present an energy reduction advantage. Due to the other properties such as the existence of micropores and the fast adsorption kinetics, it seems that the prepared nanoporous carbon has a strong potential to be used for MeO dye adsorption from aqueous media and textile wastewater. The regeneration study also demonstrated its potential to be more effectively and less expensively used.

**Table 7.** Adsorption capacities ($Q_{max}$) of MeO dye on various adsorbents.

| Adsorbents | Adsorption Capacity (mg g$^{-1}$) | Reference |
|---|---|---|
| Activated carbon/NiFe$_2$O$_4$ magnetic composite | 182.82 mg | [46] |
| Activated carbon from lignin | 300 | [47] |
| Activated carbon from date pits | 434.0 | [29] |
| Commercial activated carbon | 113.63 | [48] |
| Activated carbon Reinforced Conducting Polyaniline | 192.52 | [49] |
| Activated carbon from Lemon peels | 33 | [50] |
| Pumpkin seed powder | 200.3 | [39] |
| N-doped mesoporous activated carbon (N-OMC) | 135.8 | [51] |
| Activated carbon prepared from date pits date pits | 434 | [29] |
| Nanoporous carbon from Husks of *Moringa oleifera* | 367.83 | This Work |

## 5. Conclusions

Thermal flash pyrolysis of *Moringa oleifera* seed waste was examined for the removal of the anionic dye MeO. Pyrolysis at 900 °C and ZnCl$_2$ impregnation (1:2) were optimized to produce a high-quality nanoporous carbon ($S_{BET}$ = 699.69 m$^2$/g). The isothermal and kinetic results of MeO dye adsorption were best described by the Langmuir isotherm and pseudo-secondary reaction rate model. The adsorption capacity obtained was 367.83 mg/g. The nanoporous carbon can be recycled up to the fifth cycle to reuse its adsorption capacity. The removal efficiency of MeO dye in the current dyeing effluent is about 98%. The nanoporous carbon prepared under the mentioned conditions, $MO_{CA-ZnCl_2}$, can be considered as a

good cleaning technology for the decontamination of anionic azo dyes. This study opens several perspectives for the research and development of low-cost materials applicable to the treatment of dye effluents from the textile industry.

**Supplementary Materials:** The following supporting information can be downloaded at: https://www.mdpi.com/article/10.3390/jcs6120385/s1, Figure S1: Methyl orange dye MeO chemical structure; Table S1: Methyl orange dye MeO physicochemical characteristics; Table S2: Textural properties obtained by N2 adsorption/desorption studies on biochar MOC-ZnCl$_2$ and commercial activated carbon.

**Author Contributions:** Conceptualization, Y.R. and A.N.; Methodology, Y.R.; Validation, A.N. and S.Z.; Formal analysis, Y.R.; Resources, O.C.; Data curation, Y.R., A.N. and W.Y.; Writing-preparation of original version, Y.R.; Software, Y.R., A.N. and M.R.; Writing-reviewing and editing, Y.R., A.N., M.R., S.J.S., W.Y. and A.E.; Visualization, S.J.S.; Supervision, A.N., O.C. and S.Z. All authors have read and agreed to the published version of the manuscript.

**Funding:** This research received no external funding.

**Acknowledgments:** The authors are grateful to Omar Cherkaoui, director of research and development, for his scientific and technical contribution. In addition, they thank the management of the textile research laboratory (REMTEX) of the Higher School of Textile and Clothing Industries (ESITH) for its financial support and technical assistance. The authors also thank Issam Mechnou and Hlaibi Miloudi, members of the GEMEV laboratory team of Hassan II University, for their contributions in this work.

**Conflicts of Interest:** The authors declare no conflict of interest.

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
