# Peer review of "Efficient Adsorption of Methyl Orange on Nanoporous Carbon from Agricultural Wastes: Characterization, Kinetics, Thermodynamics, Regeneration and Adsorption Mechanism"

_jcs, doi:10.3390/jcs6120385_

Round 1
Reviewer 1 Report (Previous Reviewer 1)
The revised manuscript has greatly improved in quality compared to the original one. There is a minor typo that must be revised regarding R constant units. It reads K J/mol, and should be J/(mol K) or rather kJ/(mol K) when using 0.008314.
Apart from that, there is still something unsolved regarding the thermodynamic study. It is strange that the molecules, as stated by the authors, are less ordered after being adsorbed. If they change from a 3D to a 2d system, the order should increase. Author say that other author (Yonten) claim comparable thermodynamic parameters. They say so, but they also state a few lines above that the system is exothermic, since adsorption decreases when increasing temperature, as expected.
Authors should improve and clear up the thermodynamic study before publication
Author Response
Please see the attachment

Reviewer 2 Report (Previous Reviewer 2)
The revised version of the manuscript addresses the points identified during the review stage. The quality of the work presented and discussion improved.
Corrections needed:
1. Given the absence of an activation stage, and in line with corrections along the text that specify ZnCl2 impregnation, authors are recommended to remove the expression “activated” from the title and from the manuscript when referring to the material they developed. Instead “nanoporous carbon” can be used.
2. Correct points in line 54.
3. Table S2 - commas and points as decimals. Correct. Areas reported must not have decimal cases.
4. Regarding previous comment 4: The comment was mainly related with the use of 50 mL of water to regenerate the spent adsorbent that treated only 10 mL of solution with dye. No effective treatment of water will be achieved since during the work the dye is removed from the water to the adsorbent but to regenerate the adsorbent authors use 5 times the initial volume of water. Clarify.
5. Literature study cited for the pseudo-first and pseudo-second order kinetic models are not the original one or a reference work.
6. I cannot agree with authors answer to comment 14: The exothermic character of the adsorption process is not dependent on the physical state of the adsorbate. The overall process can benefit from the increased temperature but the adsorption step is exothermic.
7. The lixiviation of Zn must be addressed to prove that the adsorbent is lot leaching a heavy metal to the water that is being treated.
Author Response
Please see the attachment

Reviewer 3 Report (Previous Reviewer 3)
The authors addressed most of my comments and improved the manuscript by adding new information. However, some errors are still present and the overall quality of the manuscript could be further improved.
Round 2
Reviewer 1 Report (Previous Reviewer 1)
After reading the new version, I can state that authors have improved the discussion regarding thermodynamics. It seems that the process is controlled by mass transfer, reason that may explain that higher temperatures conduct to a faster difussion and therefore to a higher adsoprtion in practical equilibrium. It is still strange to me that molecules are less ordered upon adsorption, but the explanation given is well supported.
A final comment, BET and Iodine number values are enough defined with a single decimal position.
Reviewer 2 Report (Previous Reviewer 2)
Most of the issues were solved and the quality of the manuscript was improved.
The term "biochars" related with soil applications is nowadays used to describe "novel" adsorbent material obtained by carbonization and always called "chars". I believe that "chars from agricultural wastes" or "nanoporous carbons" would be more appropriate to describe the material herein presented.
Reviewer 3 Report (Previous Reviewer 3)
Even if some errors and oversights remained in the text (for example the list of publications is duplicated), the authors introduced changes in the manuscript that make the description of the experiments clearer and the discussion of the results more coherent.
In my opinion the manuscript can be now published.
This manuscript is a resubmission of an earlier submission. The following is a list of the peer review reports and author responses from that submission.
Round 1
Reviewer 1 Report
The work presented is an interesting research, but the quality of the manuscript, data presentation and discussion is below standard for a indexed publication.
See below some recommendations to improve the manuscript:
Abstract
L.22. "pseudo-order", some word (first or second" is missing
Introduction
L.35 .they (T should be capital)
L. 49. What does "high pore structure" means?
L. 51-52. Almond shell is repeated
It is stated that MeO has not been previously tested in the adsorption on this novel material, however, MeO adsorption is an standard test, that does not provide novelty by itself
Experimental
L. 99. Promimity analysis??--> Proximate analysis
L.123, 148... Formulas should be rewritten using a formula editor. Do not mix comma and dot for decimal position (use dot). This applies to many other sparts of text and tables. Check the unit homogeneity, and include conversion factors when needed
L 164. How was pH adjusted to 5?
L.172. Why isthe amount adsorbed in equilibrium described in the section devoted to kinetics?
L.192. What is the reason for sing a + symbol at the beginning of the equation?
Results
L.251. diode?-->iodine
Why are the iodine and BET results within the proximate analysis section?. It all, including proximate analysis, should be in the characterization section
Pore size distribution: the monomodal profile shown (center in the limit between micro and mesopores), does not agree with the SEM image, where much larger and irregular pores can be observed. Performing Hg porosimetry for macropores would enrich the characterization information
L.296. Micropore > 4 nm? that is not the IUPAC classification
L.299-300. The percentages are not properly written. It seems that 13% is out of the 100% of external area, not that 13% correspond to external area, out of the total
Only intraparticle diffusion model is described in the experimental section, but pseudo first and second order model are applied for discusion
What is the difference between R2 and R2 adjusted?
RL given in Table 4 (0.018) does not correspond with the given in the text (0.053), which actuaklly corresponds with KL
Reviewer 2 Report
The submitted manuscript is a classical work reporting the synthesis of activated carbons from an agricultural waste and their testing for the removal of a dye – methyl orange. The work covers the synthesis by flash carbonization followed by ZnCl2 activation, characterization of the materials and liquid phase assays to study the kinetic and equilibrium removal of methyl orange, thermodynamics, and regeneration. Thus, could fit better other journals of MDPI (e.g. C- Journal of Carbon Research) since no composite material is being reported.
Besides, the document needs deep revision from the experimental perspective and regarding the analysis of the experimental data and further discussion.
Major problems needing revision:
1. Most of the references cited in the introduction do not match with the sentences they should support.
a. Regarding dyes production (lines 34 and 35) author cite regular papers also addressing dyes removal instead of original manuscripts addressing production of dyes according to their types
b. Ref 2 is focused on activated carbons as supercapacitors but the authors cite it to support “they (dyes) are very stable and not biodegradable [2], mainly due to the presence of aromatic rings in their molecules”
c. Ref 3 and 4 are for Cu(II) and Arsenic removal but authors use it to justify “They (dyes) represent a real danger for the environment, they (dyes) are toxic substances, persistent and sometimes have a mutagenic and carcinogenic effect 39 [3] [4].”
d. Lines 51 to 53 the refs do not correspond to the listed biomasses
e. Lines 61 to 62 the refs do not correspond to the mentioned water treatment processes
f. Ref 15 in line 69 is mentioned as “our” but has no common authors with this manuscript.
2. Line 95 “Then, the activated carbon in the optimal carbonization mode was activated by the activating agent ZnCl2, at different carbon/activating agent mass ratio (1:1; 1:2; 1:3) for 3h at room temperature under a nitrogen atmosphere”. Reference literature (Marsh, H., & Rodríguez-Reinoso, F. (2006). Activated Carbon. Elsevier.) clearly shows that ZnCl2 is commonly carried out at 450-600 ºC. Please clarify and revise accordingly.
3. The presence of Zn and Cl after washing (Fig 1C, lines 274-277, Figure 4, lines 437-439) is a clear indication that the washing was not complete and Zn(II) can be leached from the activated carbon during their use for dye removal.
4. Authors use 50 mL of NaOH solution to regenerate 0.2 g of saturated materials but equilibrium studies were performed with 0.1 g of material and 10 mL of dye solution. Thus, during regeneration, the authors are generating higher volumes of solution with dye than the original ones.
5. Check the mesh on line 88 (0.250 nm), there must be an error with the units since there are no sieves with this small mesh and in section 2.6.1 it is mentioned a 1 micron filter.
6. Volume used in kinetic assays is missing.
7. Section 2.5.3: N2 adsorption data do not allow to characterize morphology, it characterized texture. Degassing conditions (time and temperature) and mass of adsorbent used must be stated. Equation 5 must be revised and a proper reference must be cited. Being a microporous material the authors must follow the recommended procedure from IUPAC (see https://doi.org/10.1515/pac-2014-1117 and https://doi.org/10.1016/S0167-2991(07)80008-5).
8. Topic 2.6.1only mentioned intraparticle diffusion but authors also report fitting to pseudo-first and pseudo-second kinetic models. Original references must be cited for each model. The same for DR equation in line 189 and BJH in line 294.
9. Topic 2.8 saturation conditions must be stated.
10. Line 265 the original paper must be cited instead of ref [3]
11. Correct errors in figure 1, legend and imagens do not correspond
12. Revise analysis of intraparticle method, point (0, 0) is missing and thus one of the linear regions. Consult for example Fig S11 and correspondent discussion of https://doi.org/10.1016/j.molliq.2019.112282.
13. Table 5: commas and points as decimals. Correct.
14. Justify the endotherm adsorption obtained when it is known that adsorption process is exothermal
15. Topic 4. Clarify the 40.12% yield. If carbonization yield is 41.36% and there was textural improvement after activation there must have been a more relevant decrease in the global yield. Besides the carbonization yield the authors must state the activation yield and the global yield that considers the total mass loss from the precursor to the final activated carbon.
16. Line 424: release of SO3- anion? There must be a mistake here.
17. Figure 4 and correspondent discussion must be revised once solved the problem with the presence of Zn(II).
18. Line 459: activated carbons are not inexpensive adsorbents. The authors must consider the actual cost of biomass, the transportation, the reagents (ZnCl2), energy, equipment…
19. Characteristics of the commercial AC and wastewater are missing and these assays must be described in the experimental section.
20. Not proofs to support sentence on lines 472/473.
Reviewer 3 Report
The manuscript concerns the adsorption of the dye methylorange by activated carbon derived from vegetal waste. I not agree with some of the considerations concerning the general role of adsorbent systems in pollutant removal. In particular for what concerns the statement at line 63-65 about possible role of adsorption in bioremediation, indeed in my opinion the removal of a pollutant by adsorption generates an high amount of polluted-adsorbent with big problems of regeneration and recovery of the final material. I invite the authors to reconsider their approach in the manuscript taking into account this point of view.
Despite this, the manuscript is an almost correct description of a series of well-defined experiments and could be publishable after some modifications and corrections.
The other drawbacks are:
- the manuscript required careful editing since many typo errors are present in text, tables and figures (i.e. “diode” instead of “iodine” at line 251, etc)
- in Figure 2C only the lines corresponding to three models are present, is the fourth line overlapped or is it missing?
- authors affirm (line 385-386) that their system is stable and do not lose affinity after repeated use. But the date in Figure 3 show that it is affected by adsorption-desorption cycles and its goodness (slowly) decrease with use. Please change this part of discussion in the manuscript.
- the statement at line 424 is not clear: which kind of coloured anion are they considering here?
- line 459: “inexpensive biosorbent”. I think this is a great mistake since the source of the material is inexpensive, but the preparation of activated carbons requires the use of energy and reagents, and also led to the dispersion of a part of the starting material. So the process is expensive and not completely environmentally sustainable. I think this aspect should be seriously considered in the manuscript.